# SCALE method for single-cell ATAC-seq analysis via latent feature extraction

Lei Xiong [1], Kui Xu[1], Kang Tian[1], Yanqiu Shao[1], Lei Tang[1], Ge Gao[2,3], Michael Zhang[4,5,6], Tao Jiang [7,8] & Qiangfeng Cliff Zhang [1]*

Single-cell ATAC-seq (scATAC-seq) profiles the chromatin accessibility landscape at single cell level, thus revealing cell-to-cell variability in gene regulation. However, the high dimensionality and sparsity of scATAC-seq data often complicate the analysis. Here, we introduce a method for analyzing scATAC-seq data, called Single-Cell ATAC-seq analysis via Latent feature Extraction (SCALE). SCALE combines a deep generative framework and a probabilistic Gaussian Mixture Model to learn latent features that accurately characterize scATAC-seq data. We validate SCALE on datasets generated on different platforms with different protocols, and having different overall data qualities. SCALE substantially outperforms the other tools in all aspects of scATAC-seq data analysis, including visualization, clustering, and denoising and imputation. Importantly, SCALE also generates interpretable features that directly link to cell populations, and can potentially reveal batch effects in scATAC-seq experiments.

---

[1] MOE Key Laboratory of Bioinformatics, Beijing Advanced Innovation Center for Structural Biology, Center for Synthetic and Systems Biology, Tsinghua-Peking Center for Life Sciences, School of Life Sciences, Tsinghua University, 100084 Beijing, China. [2] Beijing Advanced Innovation Center for Genomics (ICG), Biomedical Pioneering Innovation Center (BIOPIC), Peking University, 100871 Beijing, China. [3] State Key Laboratory of Protein and Plant Gene Research, School of Life Sciences, Center for Bioinformatics, Peking University, 100871 Beijing, China. [4] Bioinformatics Division, BNRist, Department of Automation, Tsinghua University, 100084 Beijing, China. [5] Department of Biological Sciences, Center for Systems Biology, The University of Texas, Dallas 800 West Campbell Road, RL11, Richardson, TX 75080-3021, USA. [6] MOE Key Laboratory of Bioinformatics, Center for Synthetic and Systems Biology, School of Medicine, Tsinghua University, 100084 Beijing, China. [7] Department of Computer Science and Engineering, University of California, Riverside, CA 92521, USA. [8] Bioinformatics Division, BNRIST; Department of Computer Science and Technology, Tsinghua University, 100084 Beijing, China. *email: qczhang@tsinghua.edu.cn

Accessible regions within chromatin often contain important genomic elements for transcription factor binding and gene regulation[1]. Assay for Transposase-Accessible Chromatin using sequencing (ATAC-seq) is an efficient method to probe genome-wide open chromatin sites, using the Tn5 transposase to tag them with sequencing adapters[2]. In particular, single-cell ATAC-seq (scATAC-seq) reveals chromatin-accessibility variations at the single-cell level, and can be used to uncover the mechanisms regulating cell-to-cell heterogeneity[3,4]. However, in an scATAC-seq experiment, each open chromatin site of a diploid-genome single cell only has one or two opportunities to be captured. Normally, only a few thousand distinct reads (versus many thousands of possible open positions) are obtained per cell, thus resulting in many *bona fide* open chromatin sites of the cell that lack sequencing data signals (i.e., peaks). The analysis of scATAC-seq data hence suffers from the curse of "missingness" in addition to high dimensionality[3].

Many computational approaches have been designed to tackle high-dimensional and sparse genomic sequencing data, especially single-cell RNA-seq (scRNA-Seq) data. Dimensionality reduction techniques such as principal component analysis (PCA)[5] and *t*-distributed stochastic neighbor embedding (*t*-SNE)[6] are frequently employed to map raw data into a lower dimensional space, which is particularly useful for visual inspecting the distribution of input data. Clustering based on the full expression spectrum or extracted features can be performed to identify specific cell types and states, as well as gene sets that share common biological functions[7–10]. The imputation of missing expression values is also often carried out to mitigate the loss of information caused by dropouts in scRNA-seq data[11,12].

Direct applications of these scRNA-seq analysis methods to scATAC-seq data, however, may not be suitable due to the close-to-binary nature and increased sparsity of the data (Supplementary Fig. 1). A recent method specifically developed for scATAC-seq data analysis, chromVAR[13], evaluates groups of peaks that share the same motifs or functional annotations together. Another method, scABC, weighs cells by sequencing depth and applies weighted *K*-medoid clustering to reduce the impact of missing values[14]. To refine the clustering, it then calculates a landmark for each cluster and assigns cells to the closest landmarks based on the Spearman correlation. However, each method suffers notable caveats: chromVAR only analyzes peaks in groups and lacks the resolution of individual peaks, whereas scABC heavily depends on landmark samples with high sequencing depths, and the Spearman rank can be ill-defined for data with many missing values (in particular for scATAC-seq data). Recently a newly developed method called *cis*Topic applied latent Dirichlet allocation to model on scATAC-seq data to identify *cis*-regulatory topics and simultaneously cluster cells and accessible regions based on the cell-topic and region-topic distributions[15].

Expressive deep generative models have emerged as a powerful framework to model the distribution of high-dimensional data. One of the most popular of such methods, the variational auto-encoder (VAE), estimates the data distribution and learns a latent distribution from the observed data through a recognition model (encoder) and a generative model (decoder)[16]. It does this by maximizing the similarity of the calibrated data (output by the decoder) with the input data and minimizing the Kullback-Leibler divergence of the approximate from the true posteriors[16]. VAE could be applied to data embedding and clustering based on the low-dimensional latent representation of the input high-dimensional data[17]. Recently, a method called scVI adapted VAE for scRNA-seq data analysis[18]. However, the standard VAE employed by scVI uses a single isotropic multivariable Gaussian distribution over the latent variables and often underfits sparse data[19]. A tighter estimation of the posterior distribution could greatly improve the power of VAE in fitting and analyzing sparse data[19]. Applying Gaussian Mixture Model (GMM) as the prior over the latent variables has been used in unsupervised clustering and to generate highly realistic samples by learning more disentangled and interpretable latent representations[20–22].

Here, we introduce SCALE (Single-Cell ATAC-seq analysis via Latent feature Extraction), a method that combines the VAE framework with the Gaussian Mixture Model (GMM, a probabilistic model to estimate observed data with a mixture of Gaussian distributions). We validated the effectiveness of SCALE in extracting latent features that characterize the distributions of input scATAC-seq data on multiple different datasets generated on different platforms with different protocols, and of different overall data qualities. We then used the latent features to cluster cell mixtures into subpopulations, and to denoise and impute missing values in scATAC-seq data. We compared the performance of SCALE with other widely-used dimensionality reduction techniques, as well as with the state-of-art scRNA-seq and scATAC-seq analysis tools. We found that SCALE substantially outperforms the other tools in all aspects of scATAC-seq data analysis. It is even comparable to sophisticated experimental technologies with additional steps (e.g., Pi-ATAC[23], which uses protein labeling as an aid in defining cell identifies) in correctly revealing cell types and their specific regulatory motifs in a tumor sample.

## Results

**The SCALE model and validation datasets**. SCALE combines the variational autoencoder (VAE) and the Gaussian Mixture Model (GMM) to model the distribution of high-dimensional sparse scATAC-seq data (Fig. 1). SCALE models the input scATAC-seq data $\mathbf{x}$ as a joint distribution $p(\mathbf{x}, \mathbf{z}, c)$ where $c$ is one of predefined $K$ clusters corresponding to a component of GMM, $\mathbf{z}$ is the latent variable obtained by $\mathbf{z} = \boldsymbol{\mu}_z + \boldsymbol{\sigma}_z * \boldsymbol{\epsilon}$, where $\mu_z$ and $\sigma_z$ are learned by the encoder network from $\mathbf{x}$, and $\varepsilon$ is sampled from $\mathbb{N}(0, \mathbf{I})$[16]. Since $\mathbf{z}$ is conditioned on $c$, $p(\mathbf{x}, \mathbf{z}, c)$ can be written as $p(\mathbf{x}|\mathbf{z})p(\mathbf{z}|c)p(c)$ where $p(c)$ is a discrete distribution of $K$ predefined clusters, $p(\mathbf{z}|c)$ follows a mixture of Gaussians distribution with a mean $\mu_c$ and a variance $\sigma_c$ for each component corresponding to a cluster $c$, and $p(\mathbf{x}|\mathbf{z})$ is a multivariable Bernoulli distribution modeled by the decoder network (Fig. 1). In the SCALE framework, each cell $x^i$ is first transformed into a $d$-dimensional vector of latent features $z^i$ on the GMM manifold by an encoder network and then reconstructed back through a decoder network with the original dimensionality to represent the chromatin openness at each peak in each cell. The latent features that capture the characteristics of scATAC-seq data are then visualized in the low-dimensional space with *t*-SNE, and used to cluster single cells with various clustering methods, e.g., *K*-means.

We constrained the hyper-parameters of SCALE on the Leukemia scATAC-seq dataset and found SCALE is insensitive to the encoder structure and the dimension of latent features (Supplementary Table 1). The SCALE model with default parameters can be accessed in the Online Method. We then tested the model by using the GM12878/HEK293T, the GM12878/HL-60, and the InSilico datasets[3,4,24], and two other recently published Splenocyte and Forebrain datasets[25,26]. The Leukemia dataset is derived from a mixture of monocytes (Mono) and lymphoid-primed multipotent progenitors (LMPP) isolated from a healthy human donor, and leukemia stem cells (SU070_LSC, SU353_LSC) and blast cells (SU070_Leuk, SU353_Blast) isolated from two patients with acute myeloid leukemia[24]. The GM12878/HEK293T dataset and the GM12878/HL-60 dataset are respective mixtures of two commonly-used cell lines[22]. The InSilico dataset is an in silico mixture constructed by

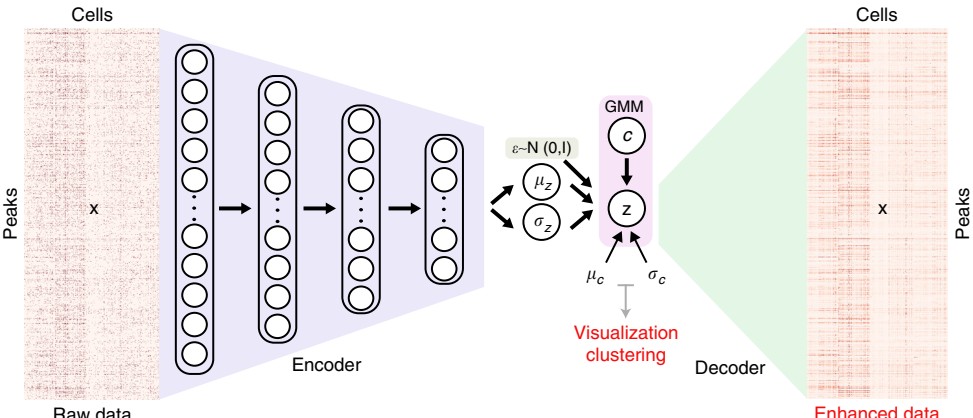

**Fig. 1** Overview of the SCALE framework. SCALE consists of an encoder and a decoder in the VAE framework. The encoder is a four-layer neural network (3200–1600–800–400) and the decoder is a network of only one layer with 10-dimensional latent variables (features) directly connected to the output. The latent variables are on the GMM manifold parameterized by $\mu_c$ and $\sigma_c$

computationally combining six individual scATAC-seq experiments that were separately performed on a different cell line[3,11]. Note that these four datasets were the same ones used to validate scABC[14]. The more recent Splenocyte dataset[25] is derived from a mixture of mouse splenocytes (after red blood cell removal) and the Forebrain dataset[26] is derived from P56 mouse forebrain cells. The six datasets cover scATAC-seq data generated from both microfluidics-based and cellular indexing platforms, and the distributions of the number of peaks in each single cell vary substantially in different datasets (Supplementary Fig. 1). However, they always have a high level of data sparsity compared to the aggregation of peaks from all single cells in each dataset (Supplementary Table. 2).

**SCALE identifies cell types by clustering on latent features**. We examined SCALE's ability to uncover features that characterize scATAC-seq data distributions. By default, SCALE extracts 10 features from the input data. For comparison, we also applied PCA, scVI and *cis*Topic to reduce the input data to 10 dimensions. In the comparison, we also included Cicero[27], a scATAC-seq data analysis tool for predicting *cis*-regulatory interactions and building single-cell trajectories from scATAC-seq data, and TF-IDF a transformation for performing dimension reduction and clustering[28]. We then visualized the extracted features from these tools as well as the raw data with *t*-SNE. In general, the feature embeddings of SCALE and *cis*Topic were better separated between cell types, whereas the embeddings of PCA, scVI, Cicero, TF-IDF and the raw data overlapped between some cell types (Fig. 2a, Supplementary Fig. 2).

SCALE can also reveal the distance between different cell subpopulations and sometimes suggested their developmental trajectory in UMAP visualization[29] (Supplementary Fig. 3). For example, in the Forebrain dataset the three clusters of excitatory neuron cells (EX1, EX2, and EX3) are close to each other in the latent space. For the Splenocyte dataset, the different T-cell subpopulations are in the neighborhood, the B cells form another bigger group, and the two types of natural killer cells also cluster closely. For the Leukemia dataset, Mono and LMPP cells are the most dissimilar in leukemia evolution and they were indeed the farthest separated. LSCs exhibit strong similarity to LMPPs[30], consistent with that the LSC cells (LSC_SU070, LSC_SU353) were close to the LMPP cells in the embedding. Finally, the blast cells (Blast_SU070 and Blast_SU353) showed a bimodal distribution, with some more differentiated blasts closer to monocytes[31,32].

We then applied *K*-means clustering on the SCALE-extracted latent features and assessed the clustering accuracy by comparing the results with scABC, scVI, *cis*Topic, and SC3[33], another widely-used clustering method for scRNA-seq data. SCALE displayed the overall best performance on all five real experimental mixture datasets, and was nearly as accurate as scABC and *cis*Topic on the InSilico dataset (Fig. 2b, Supplementary Fig. 4). The newly developed *cis*Topic generally performed pretty well on all datasets, with the overall clustering performance only slightly lower than SCALE, but it misclassified a few clusters on the Splenocyte dataset. We also compared with TF-IDF and Cicero on clustering. TF-IDF performed well on most datasets (although not as good as SCALE) except on the Forebrain dataset. However, Cicero did not perform well on most datasets; indeed, data visualization and clustering are not major goals of Cicero. On the Forebrain dataset, cluster assignments of SCALE were the closest to the reference cell types. Due to the sparsity of data, the Pearson and Spearman correlations were both ill-defined (Supplementary Fig. 5a), which directly led to poor clustering for SC3 where most cells collapsed into one group. Although the VAE-based method scVI did not suffer from the problem of ill-defined cell distance, it misclassified three subgroups of cells (s1, s2, s3 labeled on the confusion matrix. Supplementary Fig. 5b).

To identify the cause of the misclustering by scVI, we searched for the most similar cell types for the three subgroups (s1, s2, s3). We aggregated the peak profiles of each cell type or subgroup to form a representing meta-cell and calculated the similarities among the meta-cells. As expected, s1 is the most similar to EX2, s2 the most similar to EX3, and s3 to AC (astrocyte) in the original data (Supplementary Fig. 5c). Both scVI and SCALE model the distribution of scATAC peak profiles to remove noise and to impute missing values (discussed in detail in the next section). We found that, consistent with the clustering results, this data calibration by scVI actually made s1, s2, and s3 cells less similar to the original cell types of EX2, EX3, and AC, respectively. On the contrary, SCALE retained the similarities of the three subgroups to their original cell types. Strikingly, when removing the GMM restriction from the overall framework but keeping the other part of the network the same, the degenerated SCALE yield performance was similar to that of a regular VAE, like scVI (Supplementary Fig 5d). Thus, introducing GMM as the prior to restrict the data structure provides SCALE with greater power for fitting sparse data than regular VAE using single Gaussian as the prior.

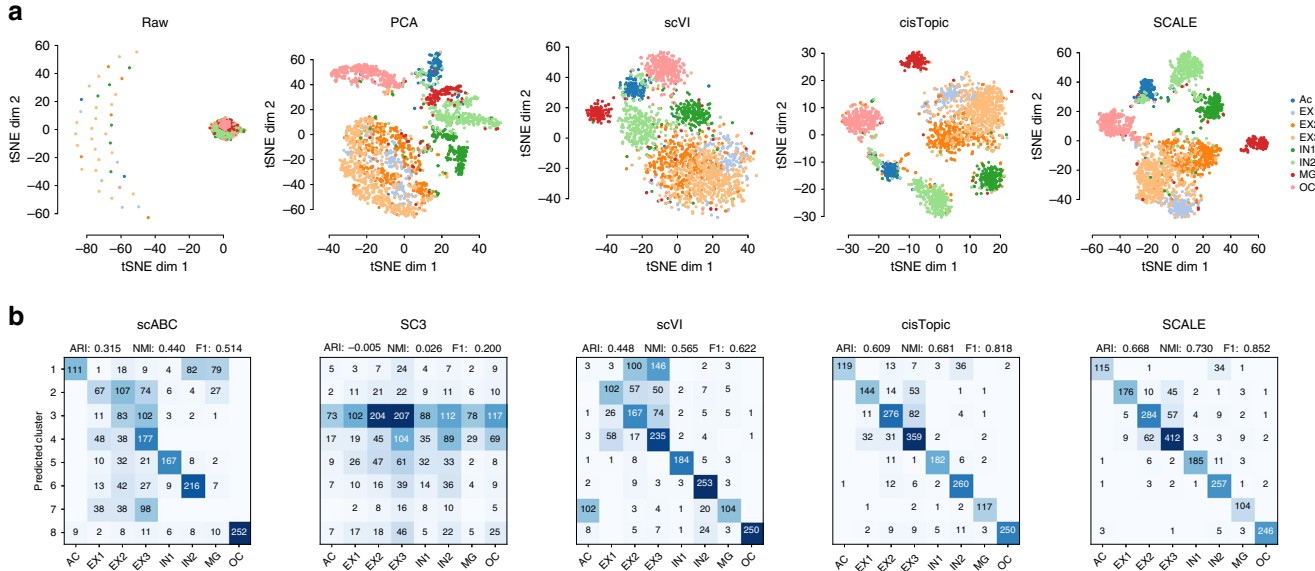

**Fig. 2** Feature embedding and clustering. **a** *t*-SNE visualization of the raw data and the extracted features from PCA, scVI, *cis*Topic, and SCALE of the Forebrain dataset. For comparison, SCALE, PCA, and scVI all performed dimension reduction to ten dimensions before applying *t*-SNE while the raw data were directly visualized with *t*-SNE. **b** Clustering accuracy was evaluated by confusion matrices between cluster assignments predicted by scABC, SC3, scVI, *cis*Topic and SCALE, and reference cell types. For scABC and SC3, the cluster assignments were directly obtained from the output of the tools; for SCALE and scVI, we applied the *K*-means clustering on the extracted features to get cluster assignments. The Adjusted Rand Index (ARI), the Normalized Mutual Information (NMI), and the F1 scores are shown on the top

Finally, we tested whether SCALE is robust with respect to data sparsity by randomly dropping scATAC-seq values in the raw datasets down to zero. We compared the clustering accuracy of SCALE and other tools at different dropping rates (10–90%), measured by the adjusted Rand Index (ARI), Normalized Mutual Information (NMI) and micro F1 score (Methods). We found that SCALE displayed only a moderate decrease in clustering accuracy with increased data corruption until at about the corruption level of 0.6, and was robust for all datasets (Supplementary Fig. 6). In general, scABC, SC3, and scVI also showed robustness to data corruption; however, the overall clustering accuracies were much lower on some datasets (e.g., SC3 failed on the Forebrain dataset and scVI failed on the GM12878/HEK293T and the GM12878/HL-60 datasets). On the Forebrain dataset, the ARI scores of SCALE dropped from 0.668 using the raw data to 0.448 on using the data with 30% corruption, and scABC and scVI dropped from 0.315 to 0.222 and from 0.448 to 0.388, respectively.

Finally we also provide a method to help users choose the optimal number of clusters based on the Tracy-Widom distribution[34] (Methods), which could often produce an estimate of the number of clusters close to that of the references (Supplementary Fig. 7) and generate clustering results similar to the reference sets (Supplementary Fig. 7).

**SCALE reduces noise and recovers missing peaks**. An important feature of SCALE is the ability to accurately estimate the real distribution of scATAC-seq data, which usually contains both noise and a large number of missing values. The estimate could be used to remove noise and restore missing data (Fig. 1). We evaluated the calibration efficiency of SCALE on both real and simulated datasets. Since no such tool is currently available for scATAC-seq data, we compared SCALE with scImpute, SAVER, MAGIC, and scVI, four state-of-the-art scRNA-seq imputation methods (Fig. 3a).

We first evaluated the ability of SCALE to remove noise and to recover missing values on real scATAC-seq datasets. A challenge

of analyzing real data is that the ground truth data without any corruption is unknown. However, if we average all single cells of the same biological cell type, the resulted meta-cell will be a good approximate to those single cells. SCALE performed better than all scRNA-seq imputation methods in all scATAC-seq datasets, in that it achieved the highest correlation of the single cells with the corresponding meta-cell for each cell type (Fig. 3a, Supplementary Fig. 8), indicating that it obtained a better estimate of the real scATAC-seq data distribution. For most cases, scImpute was very stable and among the best comparing with other scRNA-seq imputation methods, and SAVER performed well on denser datasets (InSilico, Splenocyte) but deteriorated on sparser datasets. MAGIC and scVI might have underfit the sparse input data and the imputed data substantially deviated from it (Supplementary Fig. 9), which may reflect that the two powerful tools that are optimized to scRNA-seq data imputation may not fit for scATAC-seq data analysis.

It is important to note that the data calibration of SCALE was obtained at the same time of data modeling and clustering, i.e., without knowing the original type of each cell. So it could not simply average all single cells of the same cell type to reconstruct the peak so that they resemble the reference meta-cell. Also importantly, SCALE achieved a high correlation with the meta-cells while maintaining a similar level of variation within each cell population (see the variation of correlation coefficients in Fig. 3a and Supplementary Fig. 8). Indeed, SCALE retained the original data structure (intra-correlation within the imputed data) and recovered the original peak profiles (inter-correlation with the raw data) in the process of data regularization by GMM (Supplementary Fig. 9).

The imputation of SCALE could strengthen the distinct patterns of cluster-specific peaks by filling missing values and removing potential noise (Supplementary Fig. 10), which improves downstream analysis, for example the identification of cell-type-specific motifs and transcription factors by chromVAR. We demonstrated this feature with the Forebrain dataset. We first followed the method used by Cusanovich et. al. to identify

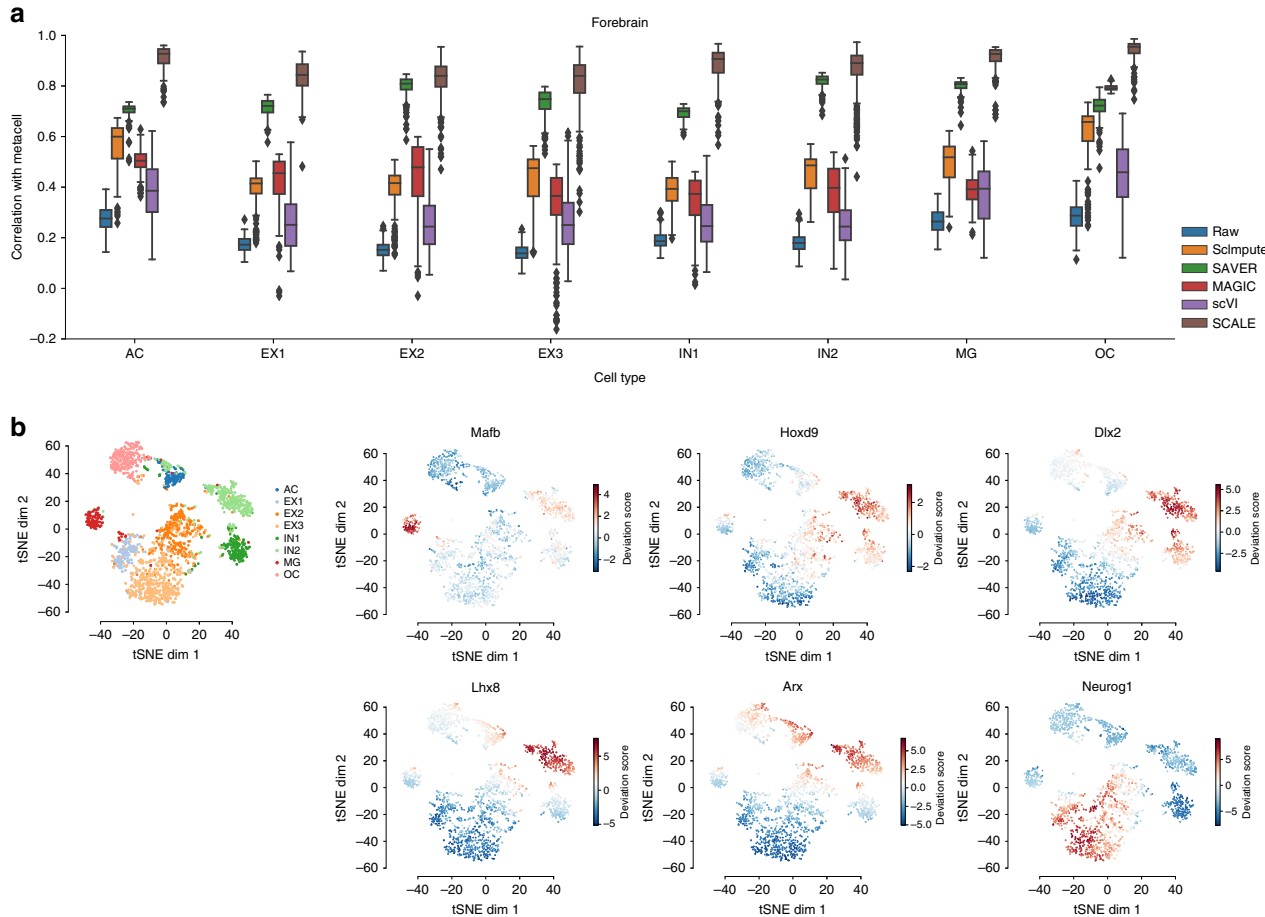

**Fig. 3** Data denoising and imputation efficiency on simulation and real datasets. **a** Comparison of the cell-wise correlations of the raw data and the calibrated data with the meta-cell of each cell type on the Forebrain dataset. **b** t-SNE embedding of 105 significant motifs profile identified by chromVAR (Benjamini-Hochberg corrected Chi-square test $p\_value\_adj$ of variability <0.05), and embeddings of motifs Mafb, Hoxd9, Dlx2, Lhx8, Arx, and Neurog1 colored by deviation scores

differentially accessible sites with the "binomialff" test of Monocle 2 package[28]. At 1% FDR threshold, we identified 4100 differential accessible sites across the eight reference clusters of the Forebrain dataset. We then used chromVAR to search for motifs enriched in the differential sites in the raw and the imputed data, respectively. Overall, the patterns of different cell types are more distinct for these differentially accessible sites in the imputed than in the raw data (Supplementary Fig. 11a). And embedding on the imputed data shows better-defined clusters (each well corresponds to a subtype with biological definition) than on the raw data (Fig. 3b, Supplementary Fig. 11b).

We found that the imputed data can greatly improve the results of chromVAR analysis by identifying more motifs (increased from 52 motifs to 105). For example, chromVAR analysis on the imputed data, but not on the raw data, identified the motifs Mafb and Hoxd9 enriched in the MG (macroglia) cluster (Supplementary Fig. 11c–d). It was recently reported that Mafb contributes to the activation of microglia[35]. It also identified Hoxd9 enriched in IN (inhibitory neuron) from the imputed but not the raw data. Similarly, we found that Dlx2, Lhx8, Arx, and Neurog1 are much more significantly enriched in the, respectively, clusters in the imputed data (Supplementary Fig. 11c-d). Dlx2, Lhx8, and Arx are important components in the MGE (medial ganglionic eminence) pathway of forebrain development[36], and Neurog1 is required for excitatory neurons in the cerebral cortex[37].

We then introduced further corruption to the real data by randomly dropping out peaks at different rates (Methods). At all

corruption rates, SCALE performed the best, in that the calibrated data most closely correlated with the original meta-cells (Supplementary Fig. 12). We observed similar trends for the other scRNA-seq imputation tools as above, confirming the effectiveness of SCALE in enhancing scATAC-seq data. We further tested the impact of missingness on generative model of imputation by calculating the confusion score (Methods) to evaluate the ability to preserve the original structure (inter and intra-correlation of meta-cells) (Supplementary Fig. 13). We found that the effect was minimal when the corruption level was lower than about 0.5, and after that threshold, the generative model was less capable of preserving the original structure (Supplementary Fig. 13b).

We subsequently tested the calibration accuracy on a simulated dataset. We constructed the dataset by first generating reference scATAC-seq data consisting of three clusters, each containing 100 peaks with no missing values, then randomly dropping out peaks and introducing noise (Methods, Supplementary Fig. 14a). As we knew the ground truth data of each single cell, we could quantify the efficiency of all tools by calculating peak-wise and cell-wise correlations of each calibrated single cell with its original ground truth. At all corruption rates, SCALE recovered the original data most accurately (Supplementary Fig. 14b–c). On the other hand, although scImpute could also recover the missing values in most cases, it messed up two clusters at the 0.2 corruption rate and was unable to remove the noise. SAVER and scVI smoothed both the signal and noise simultaneously and only recovered missing

values to some degree. MAGIC performed very well at low corruption rates, but apparently over-smoothed the data and removed true signals along with noise at high levels of data corruption.

**SCALE reveals cell types and their specific motifs**. Next, we used SCALE to analyze a dataset generated by a recently developed technology, protein-indexed single-cell assay of transposase-accessible chromatin-seq (Pi-ATAC), which uses protein labeling to help define cell identities[23]. Dissecting complex cell mixtures of in vivo biological samples may be challenging. By simultaneously characterizing protein markers and epigenetic landscapes in the same individual cells, Pi-ATAC provides an effective approach to tackle the problem. The Breast Tumor dataset is derived from a mouse breast tumor sample, including two plates of tumor cells (Epcam+) and another two plates of tumor-infiltrating immune cells (CD45+), isolated by protein labeling and FACS sorting. In the original study, a set of motifs was used to project the Epcam+ and CD45+ -specific chromatin features with t-SNE, and it was difficult to separate these two cell types computationally (Supplementary Fig. 15a). However, we found that SCALE was able to separate the two cell types well, better than PCA and scVI in latent embedding (Fig. 4a). On clustering, SCALE also yielded results the closest to the protein-index labels, better than scVI and scABC, whereas SC3 poorly distinguished the two cell types (Fig. 4b). Although cisTopic grouped the cells well in the embedding, it misclassified parts of CD45+ cells into Epcam+ cells. SCALE thus can reveal cell types within complex tissues based only on scATAC-seq data, with performance comparable to sophisticated experimental technologies like Pi-ATAC.

We validated the biological significance of the cell clusters based on Pi-ATAC peaks. For each cluster, we calculated the top 1000 peaks with the highest specificity score as type-specific peaks (Methods, Supplementary Fig. 15b). We then used Homer[38] to identify transcription factor binding motifs that were enriched in the type-specific peaks. We removed the common motifs enriched in both CD45+ cells and Epcam+ cells, and kept those that were enriched in only one cell type. We found that CD45+ cells were enriched for immune-specific motifs Maz, Pu.1-Irf, Irf8, Runx1, Elk4, Nfy, Elf3, and SpiB binding motifs. These findings are consistent with the role of Runx1 in maintenance of haematopoietic stem cells (HSC) and that knockout of Runx1 results in defective T- and B-lymphocyte development[39]. Nfy promotes the expression of the crucial immune responsive gene Major Histocompatibility Complex (MHC)[40]. Epcam+ cells were enriched for tumor-related motifs Klf14, Mitf, Ets1, Nrf2, and Nrf1 binding motifs. Ets1 is frequently overexpressed in breast cancer and associated with invasiveness[41], whereas Nrf2 is a key signature for breast cancer cell proliferation and metastasis[42] (Fig. 4c). Thus, SCALE analysis of the Breast Tumor data revealed biologically relevant cis-elements for gene regulation.

**SCALE disentangles biological cell types and batch effects**. In addition to tighter estimates of the multimodal input data, by pushing each dimension to learn a separate Gaussian distribution, GMM has another advantage in that it leads to latent representations that are more structured and disentangled, and thus more interpretable[21]. In SCALE, as each feature is directly connected with output peaks, it can be assessed by the most weighted connections (Methods, Supplementary Fig. 16a). For example, in the Leukemia dataset, dimensions 9 of the extracted features captured peaks specific to the Mono cell type and enriched regulatory elements related to immune-related "biological process" (BP, Methods) (Supplementary Fig. 16b). In the Forebrain

dataset, feature 3 characterized the AC (astrocyte) and the OC (oligodendrocyte) cell types, enriched elements related to "glial cell differentiation" (Supplementary Fig. 16c). In the Splenocyte dataset, features 4 and 7 portrayed two complementary sets of cell types (Supplementary Fig. 16d), with feature 4 enriched with B cell-related processes like "regulation of cell morphogenesis" and "myeloid leukocyte activation and differentiation", and feature 7 enriched with T cell-related processes such as "immune response" and "regulation of cell killing" (Supplementary Fig. 16d). These data suggest that the features learned by the model of SCALE are disentangled and can shed light on the biological significance.

Most interestingly, we found that SCALE could possibly reveal features corresponding to potential batch effects in the input data. For example, the Breast Tumor dataset is derived from experiments performed separately on two plates of Epcam+ tumor cell samples and two plates of CD45+ tumor-infiltrating immune cells. Although SCALE successfully clustered the two cell types, the data structure in the low-dimensional space also revealed bias towards different plates (Fig. 5a). We carefully analyzed the SCALE-extracted features (Fig. 5b) and noticed that while some, e.g., features 1 and 6, were well-correlated to biological cell types, the others, e.g., features 2, 4, 8, and 10, more or less corresponded to independent plates, or, e.g., features 3 and 5, displayed biased distribution not related to cell types. Using the plate-related features (i.e., features 2, 3, 4, 5, 8, and 10) for data embedding, we found that the cells were separated by plates, but not by types. On the other hand, if we used the other plate-independent features (i.e., features 1, 6, 7, and 9), we found that indeed the cells of different plates of the same types more evenly distributed in the cluster (Fig. 5a). We further checked the represented peaks of these features and its biological significance (Supplementary Fig. 16e). Most of plate-related features have no biological relevance, except for peaks of feature 8, which appeared in one plate of CD45+ cells and are enriched with biological processes such as "response to cytokine stimulus". This finding, however, suggests another possibility in interpreting the "plate bias" as a real biological difference in the two separate plates of CD45+ cells that might arise from sorting and cell culture.

We noticed that GM12878 cells in the InSilico dataset contain four replicates with many peak values much greater than 2. PCA analysis showed that replicates 1 and 3 were separated in the low-dimensional space (Supplementary Fig. 17a), suggesting a possible batch effect. However, the differences in the two replicates disappeared after we binarized the data, by masking values greater than 1 to 1 (Supplementary Fig. 17b). On the other hand, we observed no particular features corresponding to any batch among the SCALE-extracted features (Supplementary Fig. 17c). Consequently, in the embedding and clustering results based on the SCALE-extracted features, the cells of each replicate were distributed evenly in the low-dimensional space (Supplementary Fig. 17c). We confirmed this result by checking the top 200 specific peaks for each replicate based on raw data and found no significantly different pattern across replicates (Supplementary Fig. 17d). The distinction may reflect the different characters of the two approaches: while PCA is a linear method and sensitive to quantitative variations, SCALE is non-linear and more stable. Lastly, we repeated the analysis on the Splenocyte and the Forebrain datasets—the other two datasets that contain different experimental batches, and found no batch-related features, and the cells of different batches were distributed indistinguishably in the low-dimensional space (Supplementary Figs. 18, 19).

**SCALE is scalable to large datasets**. We further examined a mouse single-cell atlas of profiled chromatin accessibility in ~80,000 single cells from 13 adult mouse tissues by sci-ATAC-

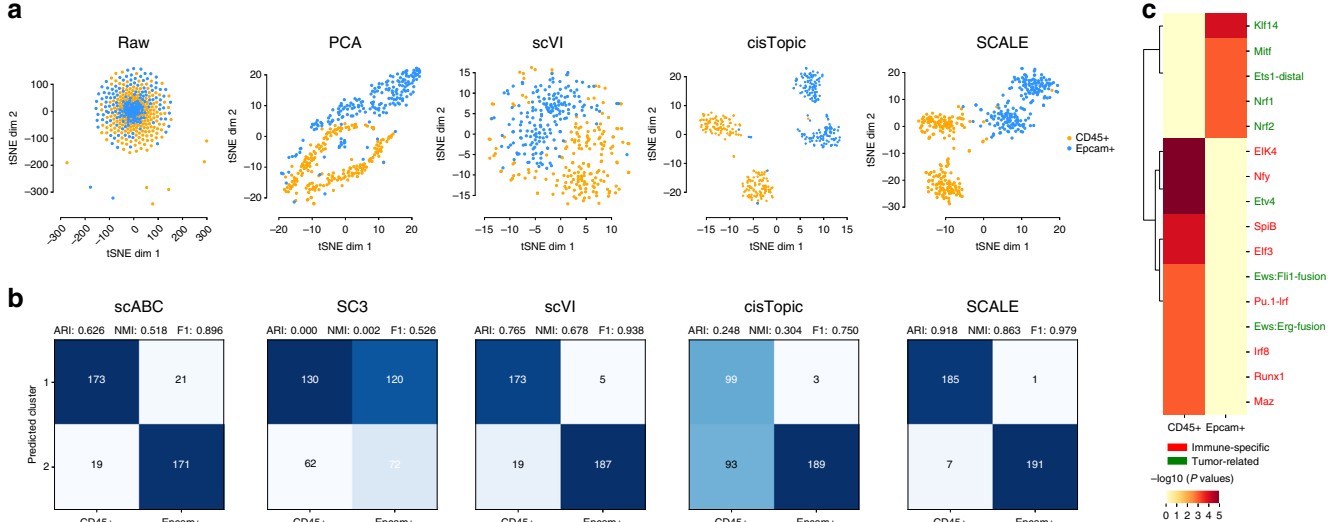

**Fig. 4** Application of SCALE on the Breast Tumor dataset from the Pi-ATAC study. **a** *t*-SNE visualization of the Breast Tumor raw data, and features extracted by PCA, scVI, and SCALE. **b** clustering results by scABC, SC3, scVI, *cis*Topic, and SCALE. **c** Heatmaps of enriched motifs of different transcription factors across CD45+ cells and Epcam+ cells from the mouse breast tumor sample

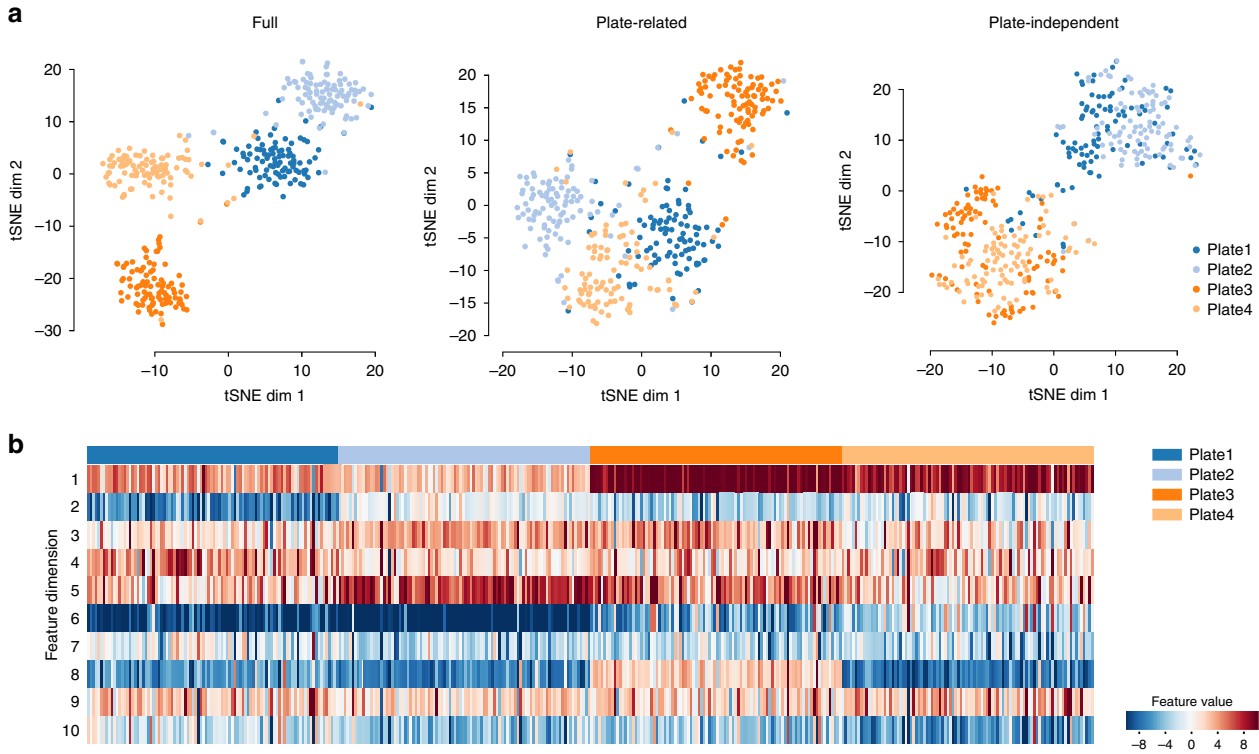

**Fig. 5** Batch effects and biological cell types are captured by disentangled latent features. **a** *t*-SNE visualization of full features, plate-related features (features 2, 3, 4, 5, 8, 10), and plate-independent features (features 1, 6, 7, 9). **b** Heatmap of the ten features learned by SCALE. Different batches (plates) are colored by different colors on the top, plate 1 and 2 are Epcam+ cells, plate 3 and 4 are CD45+ cells

seq[28] to investigate whether SCALE works for large datasets. The atlas study used a computational pipeline to infer 30 cell types from the dataset by graphic clustering, which were used as "reference" cell types when benchmarking SCALE. SCALE worked well on this big dataset and showed a good agreement with the reference: the overall F1 score was 0.419, and most of the major reference clusters have a corresponding one identified by SCALE. Nevertheless, some large reference clusters were split into two or three small groups (Supplementary Fig. 20).

Finally, we benchmarked the running time and memory usage of SCALE on different scales of datasets by downsampling a subset of cells and peaks from the mouse atlas datasets (10,000 peaks and different cell number). We found that SCALE required a little over 1.5 h and 2 GB of memory to run a dataset, and

importantly the used computational resource only increased slightly with the size of datasets (Supplementary Fig. 21).

## Discussion

Our work shows that SCALE accurately characterizes the distribution of high-dimensional and sparse scATAC-seq data by using a deep generative framework to extract latent features. SCALE is thus a powerful tool for scATAC-seq data analysis, including data visualization, clustering, and denoising and imputation. In all comparisons, SCALE performs much more favorably than scABC and scRNA-seq tools. Based on the better clustering assignments and imputation data, we can improve the discovery of cluster-specific peaks, and regulatory motifs as well, when combined with tools like Homer[38] or chromVAR[13].

The success of SCALE can be attributed to the powerful deep generative framework and the GMM to accurately model the high-dimensional, sparse, multimodal scATAC-seq data. Similar to SCALE, a recent scRNA-seq analysis tool scVI also learns latent representation of scRNA-seq data by aggregating information across similar cells using a hierarchical Bayesian model[18]. However, SCALE also applies a GMM to overcome the increased sparsity of scATAC-seq data and more tightly estimate data distribution, thus achieving higher accuracy than scVI on scATAC-seq data analysis. It highlights the necessity and advantage to develop new methods that are optimized for scATAC-seq data, but not to use scRNA-seq data analysis tools.

An attractive additional observation about SCALE is the interpretability of the GMM model. We showed that SCALE could possibly capture biological cell-type-related and potential batch-effect-related latent features in the low-dimensional space. By excluding batch-related features in embedding and clustering, we are able to reduce batch effects. Nevertheless, SCALE is not specifically designed to identify and remove these artifacts from the input data. In the future, we could improve the model to explicitly incorporate variables that are designated for the discovery and removal of batch effects and other possible technical variations.

## Methods

**Data and preprocessing**. *Data*: The Leukemia dataset is derived from a mixture of monocytes (Mono) and lymphoid-primed multipotent progenitors (LMPP) isolated from a healthy human donor, and leukemia stem cells (SU070_LSC, SU353_LSC) and blast cells (SU070_Leuk, SU353_Blast) isolated from two patients with acute myeloid leukemia[24]. The GM12878/HEK293T dataset and the GM12878/HL-60 dataset are respective mixtures of two commonly-used cell lines[3]. The InSilico dataset is an in silico dataset constructed by computationally putting together six individual scATAC-seq experiments separately performed on a different cell line[3,11]. The Splenocyte dataset[25] is derived from a mixture of mouse splenocytes (after red blood cells removal) and the Forebrain dataset[26] is derived from P56 mouse forebrain cells. The Breast Tumor dataset[23] is obtained from a mouse breast tumor sample, including two plates of tumor cells (Epcam+) and another two plates of tumor-infiltrating immune cells (CD45+) from protein labeling and FACS sorting.

*Preprocessing*: Similar to scABC[14], we filtered the scATAC-seq count matrix to only keep peaks in10 cells with ≥2 reads for the InSilico dataset, the GM12878/HEK293T dataset, and the GM12878/HL-60 dataset, ≥5 cells with ≥2 reads for the Leukemia dataset, ≥50 cells with ≥2 reads for the Forebrain dataset, and ≥5 cells with ≥1 reads for the Breast Tumor dataset. We kept all the peaks for the Splenocyte dataset. We also only kept cells with read counts ≥(number of filtered peaks/50). For the InSilico dataset, there were still almost 90,000 peaks after filtering. For the efficiency of the SCALE model, similar to SC3[33], we further removed rare peaks (reads >2 in less than X% of cells) and ubiquitous peaks (reads ≥1 in at least (100−X)% of cells).

**The probabilistic model of SCALE**. SCALE combines a variational autoencoder (VAE) and the Gaussian Mixture Model (GMM) to model the input scATAC-seq data $\mathbf{x}$ through a generative process. Given K clusters, corresponding latent variable $\mathbf{z}$ can be obtained through the encoder via the reparameterization then to generate sample $\mathbf{x}$ through the decoder. It can be modeled with a joint distribution $p(\mathbf{x}, \mathbf{z}, c)$, where $\mathbf{z}$ is the latent variable and $c$ is a categorical variable whose probability is Discrete $(c|\pi)$ where $P(C = j) = \pi_j$, $\pi \in \mathbb{R}^K$. $p(\mathbf{z}|c)$ is mixture of Gaussians distribution parameterized by $\mu_c$ and $\sigma_c$ conditioned on $c$. Given that $\mathbf{x}$ and $c$ are

independently conditioned on $\mathbf{z}$, then joint probability $p(\mathbf{x}, \mathbf{z}, c)$ can be factorized as:

$$p(\mathbf{x}, \mathbf{z}, c) = p(\mathbf{x}|\mathbf{z})p(\mathbf{z}|c)p(c) \tag{1}$$

We define each factorized probability as:

$$p(c) = \text{Discrete}(c|\pi) \tag{2}$$

$$p(\mathbf{z}|c) = \mathbb{N}(\mathbf{z}|\mu_c, \sigma_c^2 \mathbf{I}) \tag{3}$$

$$p(\mathbf{x}|\mathbf{z}) = \text{Ber}(\mathbf{x}|\mu_x) \tag{4}$$

The training SCALE is to maximize the log-likelihood of the observed scATAC-seq data:

$$\log p(\mathbf{x}) = \log \int_z \sum_c p(x, z, c)dz \tag{5}$$

$$\geq E_{q(\mathbf{z}, c|\mathbf{x})}\left[\log \frac{p(\mathbf{x}, \mathbf{z}, c)}{q(\mathbf{z}, c|\mathbf{x})}\right] = \mathcal{L}_{\text{ELBO}}(\mathbf{x}) \tag{6}$$

which can be transformed to maximize the evidence lower bound (ELBO). The ELBO can be written with a reconstruction term and a regularization term:

$$\mathcal{L}_{\text{ELBO}}(\mathbf{x}) = E_{q(\mathbf{z}, c|\mathbf{x})}[\log p(\mathbf{x}|\mathbf{z})] - D_{KL}(q(z, c|\mathbf{x})||p(z, c)) \tag{7}$$

The reconstruction term encourages the imputed data to be similar to the input data. The regularization term is a Kullback–Leibeler divergence, which regularizes the latent variable $\mathbf{z}$ to a GMM manifold. And $q(\mathbf{z}, c|\mathbf{x})$ and $p(\mathbf{x}|\mathbf{z})$ are an encoder and a decoder, respectively, which can be modeled by two neural networks.

**The overall network architecture of SCALE**. SCALE consists of an encoder and a decoder. The encoder is a four-layer neural network (3200–1600–800–400) with the ReLU activation function. The decoder has no hidden layer but directly connects the ten latent variables (features) to the output layer (peaks) with the Sigmoid activation function. A GMM model is used to initialize the parameters $\mu_c$ and $\sigma_c$. The Adam optimizer[43] with a 5e-4 weight decay is used to maximize the ELBO. Mini-batch size is 32. SCALE also provides a quick mode for large datasets with the encoder structure of two layers (1024–128), and model training with maximum iterations of 30,000, and early stopping when no improvements in 10 epochs. The GMM models are constructed with the Python "scikit-learn" package, and the neural network is implemented with the "pytorch" package.

**Visualization**. We used $t$-SNE from the Python "scikit-learn" package to project the raw data or latent features to 2-dimension with random state as 124. We used Python package "umap" to visualize the trajectory cell relationships.

**Clustering**. We used the $K$-means clustering method from the Python "scikit-learn" package to cluster the input single cells based on the extracted features. To repeat the result, we set the random seed to 18.

**Evaluation of clustering results**. *Adjusted Rand Index*: The Rand Index (RI) computes similarity score between two clustering assignments by considering matched and unmatched assignments pairs independently of the number of clusters. The Adjusted Rand Index (ARI) score is calculated by "adjust for chance" with RI by:

$$\text{ARI} = \frac{RI - \text{Expected\_RI}}{\max(RI) - \text{Expected\_RI}}$$

If given the contingency table, the ARI can also be represented by:

$$\text{ARI} = \frac{\sum_{ij} \binom{n_{ij}}{2} - \left[\sum_i \binom{a_i}{2} \sum_j \binom{b_j}{2}\right] / \binom{n}{2}}{\frac{1}{2}\left[\sum_i \binom{a_i}{2} + \sum_j \binom{b_j}{2}\right] - \left[\sum_i \binom{a_i}{2} \sum_j \binom{b_j}{2}\right] / \binom{n}{2}}$$

The ARI score is 0 for random labeling and 1 for perfectly matching. *Normalized mutual information*:

$$\text{NMI} = \frac{I(P, T)}{\sqrt{H(P)H(T)}}$$

where P, T are empirical categorical distributions for the predicted and real clustering, $I$ is the mutual entropy, and $H$ is the Shannon entropy. *F1 score*:

$$\text{score} = 2 * (\text{precision} * \text{recall})/(\text{precision} + \text{recall})$$

**Generation and corruption of the simulation dataset**. A simulation dataset consisting of 300 cells and 1000 peaks was generated. The peaks formed three clusters, with each cluster containing 100 specific peaks. These specific peaks had a value of 1 or 2 (ratio 1:4) in the cells of the corresponding clusters, and 0 in other cells. Corrupted datasets were generated by randomly dropping out values at different rates from 0.1 to 0.8, followed by introducing random noise by setting values as 1 or 2 (ratio 1:4) with the probability of 0.1.

**Identifying differentially accessible sites**. We followed Cusanovich et al. [28] and used "binomiallf" test implemented in Monocle 2 package[44] to identify differentially accessible peaks. We set a 1% FDR threshold (Benjamini-Hochberg method) to decide the peaks were significant for each cluster.

**Calculation of the cluster specificity score of a peak**. We applied an entropy-based measure to calculate a cluster specificity score for the association of each peak with each cluster. In detail, it is defined by comparing the distribution of the peak pattern with the predefined ideal cluster-specific pattern in which a peak only appears in one cluster:

$$\text{score} = 1 - \sqrt{\text{Div}_{\text{jensen}}(p, q)}$$

while $p$ is the distribution of observed peaks overall samples, and $q$ is the distribution of predefined pattern for the cluster $c$,

$$q = \left(q_1^{c_1}, q_2^{c_2}, \ldots, q_n^{c_n}\right) s.t\ q_i^{c_i} = \begin{cases} 1, & \text{if } c_1 = c \\ 0, & \text{else} \end{cases}$$

where $\text{Div}_{\text{jensen}}(p, q)$ is the Jensen divergence distance:

$$\text{Div}_{\text{Jensen}}(p, q) = H\left(\frac{p + q}{2}\right) - \frac{H(p) + H(q)}{2}$$

where $H(p)$ is the entropy of peak's distribution:

$$H(p) = -\sum_{i=1}^{n} p_i \log(p_i)$$

This provides the peak-cluster matrix, and the final cluster specificity score is the maximal score overall clusters. By default, we defined the top 200 peaks as the cluster-specific peaks, which were used in the downstream analysis.

**Binarization**. We transformed the float imputed values to binary ones as below:

$$\text{imputed}_{i,j} = \begin{cases} 1, & \text{if imputed}_{i,j} > \text{mean}(\text{raw}_{i,:}) \text{ and } > \text{mean}(\text{raw}_{:,j}) \\ 0, & \text{else} \end{cases}$$

where imputed is the imputation matrix, raw is the raw data matrix, $i$ means the $i$th peak, $j$ means the $j$th cell.

**Confusion score**. We first calculated the inter/intra-correlation matrix, then transformed the diagonal values of the correlation matrix to:

$$\text{Correlation}_{\text{diag}} = 1 - \text{Correlation}_{\text{diag}}$$

Then calculated the mean of the upper triangle of the correlation matrix as the confusion matrix:

$$\text{confusion score} = \text{mean}(\text{Correlation}_{\text{triu}})$$

A confusion score of "0" means a perfect preservation of the original population.

**Features associated peaks**. In SCALE, as each feature is directly connected with output peaks, the feature-peak association can be assessed by the weights of links. We approximate the distribution of the weights as a Gaussian distribution, and defined those peaks with weights most deviated from the mean as feature-associated peaks. By default, we set 2.5 standard deviations from the mean as the cutoff.

**Discovery of enriched TFs**. We applied *findMotifsGenomes.pl* from the software Homer with default parameters on the top 1000 specific peaks of the CD45+ and the Epcam+ corresponding single-cell clusters, respectively, to search for transcription factor binding motifs. We only considered the motif occurrences with binomial test $P$-value $\le 0.001$.

**Annotation of genomic elements**. We used the GREAT[45] algorithm (version 3.0.0) to perform the gene enrichment analysis by including genomic regions of a basal plus an extension (1 kb upstream and 0.1 kb downstream with up to 500-kb max extension) in the search for elements enriched with the GO 'biological process' terms.

**Prediction of a suitable number of cluster k**. We used the number of the eigenvalues of $X^TX$ that are significantly different as the predicted k, where $X$ is the count matrix. We followed SC3 and calculated the mean and the s.d. of the Tracy-Widom distribution to determine the threshold:

$$\text{mean} = \left(\sqrt{n-1} + \sqrt{p}\right)^2$$

$$\text{s.d.} = \left(\sqrt{n-1} + \sqrt{p}\right)\left(\frac{1}{\sqrt{n-1}} + \frac{1}{p}\right)^{\frac{1}{3}}$$

Where n is the number of peaks and $p$ is the number of cells.

**Reporting summary**. Further information on research design is available in the Nature Research Reporting Summary linked to this article.

## Data availability

The scATAC-seq in silico mixture data are available in Gene Expression Omnibus (GEO) under accession number GSE65360. Single-cell data for leukemia mixture is available at GSE74310. GM12878/HEK293T and GM12878/HL-60 mixtures can be found at GSE68103, Pi-ATAC Breast Tumor data can be obtained at GSE112091. Splenocyte mixture can be accessed at ArrayExpress with accession number E-MTAB-6714 and Forebrain mixture can be accessed at GSE100033. The mouse atlas dataset is available at http://atlas.gs.washington.edu/mouse-atac. All other relevant data supporting the key findings of this study are available within the article and its Supplementary Information files or from the corresponding author upon reasonable request. A reporting summary for this Article is available as a Supplementary Information file.

## Code availability

The SCALE software including documents and tutorial is available on Github (https://github.com/jsxlei/SCALE).

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

## Acknowledgements

We thank Xinqi Chen for insightful comments on the manuscript and the help with the investigation of the Breast Tumor dataset. We thank Jianbin Wang for helpful suggestions. We thank Mahdi Zamanighomi and Timothy Daley for kindly providing the InSilico and the Leukemia datasets used in the scABC paper. We thank Rongxin Fang for the cell-type labels of the Forebrain dataset in their original paper. We thank Life Science Editors for editing assistance. This project is supported by the Chinese Ministry of Science and Technology (Grant No. 2018YFA0107603 to Q.C.Z.) and the National Natural Science Foundation of China (Grants No. 91740204, 31761163007, and 31621063 to Q.C.Z.), the Beijing Advanced Innovation Center for Structural Biology, the Tsinghua-Peking Joint Center for Life Sciences and the National Thousand Young Talents Program of China to Q.C.Z.

## Author contributions

Q.C.Z. conceived and supervised the project. L.X. designed, implemented, and validated SCALE with the help from K.X., K.T. and Y.S., L.T., G.G., M.Z. and T.J. helped analyzing the data, L.X. and Q.C.Z. wrote the manuscript with inputs from all the authors.

## Competing interests

The authors declare no competing interests.
