## [Peer Review File · Nature Communications]

Reviewers' comments:

Reviewer #1 (Remarks to the Author):

Single-cell ATAC-seq data is extremely sparse, which presents major difficulties in its analysis. Until now, there are only a few methods that address this problem. In this work, Xiong et. al. describe a new method for dealing with the sparsity of single-cell ATAC-seq data using latent feature extraction. The authors demonstrate the usefulness of the method in visualizing, clustering and imputing in several scATAC-seq datasets. In particular, the success in visualization and clustering would be useful to the community, as would the potential interpretability of the extracted features. However, the authors need to address some outstanding issues and are lacking comparisons to some existing methods:

1. Figure S1: What is being plotted in part B? The caption reads 'number of cells with non-zero peaks'. Should this be the number of peaks with non-zero counts per cell? If not, then some kind of reads per cell metric should be shown.
2. tSNE is not designed to maintain relationships at large distances, but only at close distances. The authors should not infer trajectory-type relationships using tSNE (lines 145-155).
3. The authors should consider some of the clustering and visualization methods currently in use. The authors should either compare performance of these methods or discuss how the goals of SCALE are different. In particular, the methods used in
 - a. Cusanovich et. al. Cell, 2018 <https://doi.org/10.1016/j.cell.2018.06.052> (TF-IDF > SVD > tSNE)
 - b. Cicero <https://cole-trapnell-lab.github.io/cicero-release> from Pliner et.al. Molecular Cell 2018
4. The authors do not discuss or demonstrate the benefits of imputation for scATAC-seq. Is there any gain in biological interpretability i.e. identification of differential accessibility, further subtyping of cells, etc?
5. The authors discuss the interpretability of the SCALE features beginning on line 272, however they do not discuss what the sets of chosen peaks are, for example, it would be interesting to see a gene set enrichment analysis of the nearest (or predicted) genes of the peaks that make up a feature. A similar analysis of the batch effect peaks would also be interesting.
6. Newer scATAC-seq datasets often have median reads per cell in the 10s of thousands. The authors use one dataset of similar density (splenocyte), but I would be interested in seeing another example, especially as the splenocyte dataset most often had similar performance with scVI. An example of denser datasets available are those published by 10X Genomics from their new platform.
7. Figure S5: I am curious about the limits of the SCALE method to downsampling. The authors should continue to downsample the data beyond 0.3 (perhaps plotted using line charts rather than barcharts). It would also be useful to use downsampling to investigate the relationship between performance and reads per cell, for example by plotting the number of reads per cell after downsampling compared to performance.
8. Was any tuning required to run SCALE on the different datasets provided? Were any non-default options used?

9. Lastly, the authors should make some comment about the computational performance of the method – even just a note on whether it takes seconds, minutes, or hours to run on some moderately sized dataset.

Minor points:

Line 47: Cusanovich et al. Science 2015 should also be cited along with Buenrostro for the invention of single-cell ATAC-seq.

Line 138: Identifies cell types, does not discover them.

Figure S3: The authors should remove the bottom rows from the confusion matrices in rows 2 and 3.

In several figures, the authors use a three-color scale (red to white to blue) when displaying values that do not have a natural mid-point (i.e. all are positive or negative, none pass zero – the white midpoint has no special meaning). The figures would be easier to decipher on a two-color scale.

Reviewer #2 (Remarks to the Author):

In this manuscript Lei Xiong et al. applied a variational autoencoder combining with Gaussian Mixture Model to encode latent features that could accurately characterize single cell ATAC-seq data. scATAC-seq is binary, sparse and noisy. So it is difficult to directly apply analysis methods developed for bulk ATAC-seq data. Also methods developed for scRNA-seq cannot be directly applied to analyze scATAC-seq due to the nearly binary nature of the later. Lei Xiong applied variational encoder which learns a latent distribution of the observed data through an encoder (recognition model) and the decoder (generative model). However, regular VAE often underfits sparse data. So they introduced Gaussian Mixture Model (GMM) as a prior over the latent variables that gives more power in fitting sparse data than regular VAE. Overall this is an important work given how few methods/tools are available that supports scATAC-seq analysis. However, I feel that some more work is needed before I can recommend it for publication.

Comments:

1. Although the authors have benchmarked the tool with number of published datasets, the number of single cells were moderate (the maximum was 3166 cells). As, 10X and ICELL8 are now launching their scATAC-seq protocols it would soon be common to have scATAC-seq applied to large number of single cells. So I would like to see how the tool scales to large datasets. One such dataset could be of Cusanovich et. al. Cell (2018) DOI:<https://doi.org/10.1016/j.cell.2018.06.052> with ~100,000 single cells. Even if it is not possible to work with all the cells, it would be good to know how many cells SCALE can handle and how the time/memory scales with increasing number of cells.

Also, in that paper Cusanovich reported identification of 30 major clusters which were then broken down to 85 clusters in total. How many of these clusters can SCALE pick?

2. It is mentioned in the paper that SCALE is insensitvie to the encoder structure and the dimension of latent features. But no benchmarking results based on different structure and dimensions of latent spaces were presented. Presenting this data in the paper would be very useful.

3. It is always challenging to predict the number of clusters 'K' in the model. Users can run GMM separately and use BIC or other such measures to identify possible number of clusters. But an embedded approach with SCALE to predict the 'K' would be easier for the user to run SCALE all at one go.

4. When there would be minor clusters along with major clusters, for eg. with Cusanovich dataset, which one should be used for the GMM?

5. The imputed data from SCALE is floating point values, but the scATAC-seq is effectively binary (counts of more than one would represent other alleles of a locus, but this would be rare) value. Downstream algorithms that applies binary assumption of the data would not work on this imputed data. How would the user address this issue? Also, the author mentions in the paper that SCALE recovered the original peak profiles in the process of data regularization by GMM. But will it not be the case that the original peak profiles would be of binary value?

6. The authors showed that SCALE could identify features that are related to batch effects from the input data in their Breast Tumor dataset. In the GM12878 cell line data, PCA analysis was separating cells based on replicates. But the figure they are referring to are t-SNE plots and I could not find how many PCAs were taken as input to generate this t-SNE plot. Also, which Principle Component was responsible for the batch effect.

SCALE could not detect any batch effect in GM12878 cell line dataset. The authors argued that the batch effect might not be strong enough to come up as a latent space feature and thus was not detected as a latent space. But this seems to be ambiguous to me. If SCALE has the capability of identifying batch effect it should do so even when the batch effect signal is not strong. The user can then work around this feature to take in to consideration for further downstream analysis.

7. What specific attributes of the dataset causes this differences in identifying batch effect, is it the sparsity of the data or heterogeneity in the data or something else?

8. I tried the tool in one of our local dataset where the data for the peaks has been binarized (any value more than 1 has been made 1 although they were few). In the first Epoch it gave a loss value and lr, but after that all the epoch produced nan and ended with error. This needs to be fixed.

Reviewer #3 (Remarks to the Author):

This paper proposes to use variational autoencoders (VAE) with a Gaussian mixture model (GMM) latent prior distribution for analysis of scATAC-seq data (Single-cell profiles of chromatin-accessibility landscape at single cellular level). This is a timely contribution, well-written with a thorough analysis of a number of datasets. A number of comments focussing on the modelling part are given in the following:

Major comments:

1. The Gaussian mixture model (GMM) variational autoencoder (VAE) model has been applied to scRNA-Seq previously <https://www.biorxiv.org/content/10.1101/318295v2> This does not invalidate this paper because its main focus is ATAC-seq but the paper is relevant to cite.

2. K-means clustering is applied to clustering of the learned latent features. Why not use the responsibilities (soft cluster assignments) that the algorithm gives. The GMM is a clustering algorithm by itself.

3. The ATAC data is sparse from missing data as stated in the paper: "Normally, only a few thousand distinct reads (versus many thousands of possible open positions) are obtained per cell, thus resulting in many bona fide open chromatin sites of the cell without sequencing data signals (i.e., peaks)." It could be good with a discussion about what is the adequate model of missingness and how that affects the formulation of the generative model. It could be the one proposed. But more justification is needed.

Minor comments:

1. The analysis of scATACseq data hence suffers from the curse of sparsity.. Perhaps it is more accurate to say

The analysis of scATACseq data hence suffers from the curse of missingness

2. The Seurat method <https://satijalab.org/seurat/> seems to emerge as a standard method for scRNA-Seq analysis. Could be good to see a comparison to Seurat.

3. Good to see the code is available!

Reviewer #1 (Remarks to the Author):

Single-cell ATAC-seq data is extremely sparse, which presents major difficulties in its analysis. Until now, there are only a few methods that address this problem. In this work, Xiong et. al. describe a new method for dealing with the sparsity of single-cell ATAC-seq data using latent feature extraction. The authors demonstrate the usefulness of the method in visualizing, clustering and imputing in several scATAC-seq datasets. In particular, the success in visualization and clustering would be useful to the community, as would the potential interpretability of the extracted features. However, the authors need to address some outstanding issues and are lacking comparisons to some existing methods:

RESPONSE: We thank the reviewer for recognizing the significance of our work and also the thoughtful comments. We have addressed all the issues, below. In particular, we provided additional comparisons with existing methods. The results show that SCALE outperforms the others in all aspects. Finally we improved the manuscript, with revised text, figures, tables highlighted in yellow.

1. Figure S1: What is being plotted in part B? The caption reads 'number of cells with non-zero peaks'. Should this be the number of peaks with non-zero counts per cell? If not, then some kind of reads per cell metric should be shown.

RESPONSE: We apologize for the confusion. The reviewer is right, we have changed the caption to "number of peaks with non-zero counts" (**Supplementary Figure S1**).

2. tSNE is not designed to maintain relationships at large distances, but only at close distances. The authors should not infer trajectory-type relationships using tSNE (lines 145-155).

RESPONSE: We appreciate the reviewer for pointing out the inappropriate use of tSNE in the inference of the trajectory of cancer development. We now used UMAP, a global distance-preserving method that is frequently used for trajectory analysis. The results are similar, and support the original conclusions (see the revised text at **line 151**). Please see the new **Supplementary Figure S3**.

3. The authors should consider some of the clustering and visualization methods

currently in use. The authors should either compare performance of these methods or discuss how the goals of SCALE are different. In particular, the methods used in

a. Cusanovich et. al. Cell, 2018 <https://doi.org/10.1016/j.cell.2018.06.052> (TF-IDF > SVD >tSNE)

b. Cicero <https://cole-trapnell-lab.github.io/cicero-release> from Pliner et.al. Molecular Cell 2018

RESPONSE: We thank the reviewer for his/her suggestion to include more clustering and visualization for comparison. We have compared both TF-IDF and Cicero for scATAC-seq data visualization and clustering (line 144, the new **Supplementary Figure S2** and **Figure S4**). Overall, both TF-IDF and Cicero could separate different cell types and preserve their relative relationships for most datasets. However, TF-IDF mixed the Forebrain dataset and did not separate well the Leukemia and the GM12878/HL-60 datasets, and Cicero also mixed the Forebrain dataset to some extent. These observations are also reflected in their clustering performance as discussed below in detail.

For clustering by TF-IDF, we followed the tutorial given at <http://atlas.gs.washington.edu/fly-atac/docs/#use-case-2-re-cluster-cells-with-t-sne>. For consistency, we kept the same number of clusters as the reference labels by adjusting the *delta* and *rho* parameters in the density peak algorithm used by TF-IDF for clustering. For Cicero, we followed the manual at <https://cole-trapnell-lab.github.io/cicero-release/docs/#constructing-single-cell-trajectories>. As can be seen from the new **Supplementary Figure S4**, TF-IDF performed well on most datasets (although not as good as SCALE) except on the Forebrain dataset, whereas Cicero does not perform well on most datasets. It is worthy to note that this finding might reflect the major goal of Cicero, which is to predict cis-regulatory interactions and to build single-cell trajectories from scATAC-seq data, not to visualize and cluster data.

In addition to TF-IDF and Cicero, we also compared SCALE with another newly released software tool, cisTopic, for data visualization and clustering (line 143, 163, new **Supplementary Figure S2** and **Figure S4**). As can be seen from the figure, cisTopic performed pretty well on all datasets, with the overall clustering performance only slightly lower than SCALE. But it misclassified a few clusters on the Splenocyte dataset.

4. The authors do not discuss or demonstrate the benefits of imputation for scATAC-seq. Is there any gain in biological interpretability i.e. identification of differential accessibility, further subtyping of cells, etc?

RESPONSE: We especially thank the reviewer for the suggestion to discuss the benefits of imputation for scATAC-seq. In general, we found that downstream analysis, e.g., the discovery of cell-type-specific motifs and transcription factors, could be improved after data imputation.

We demonstrated this with the Forebrain dataset. We first followed the method used by *Cusanovich et. al.* to identify differentially accessible sites with the “binomialff” test of Monocle 2 package¹. At 1% FDR threshold, we identified 4100 differential accessible sites across the 8 reference clusters of the Forebrain dataset. We then used chromVAR to search for motifs enriched in the differential sites in the raw and the imputed data respectively (p_value_adj of variability < 0.05).

The new **Supplementary Figure S11** shows the heatmaps of scATAC-seq peak profiles, tSNE embeddings of the whole dataset, as well as the enrichment of individual transcription factors in each cell. Overall, the patterns of different cell types are more distinct for these differentially accessible sites in the imputed than in the raw data (**panel a**). And embedding on the imputed data shows better-defined clusters (each well corresponds to a subtype with biological definition) than on the raw data (**panel b**).

We can see that the imputed data can greatly improve the results of chromVAR analysis by identifying more motifs (increased from 52 motifs to 105). For example, chromVAR analysis on the imputed data, but not on the raw data, identified the motifs *Mafb* and *Hoxd9* enriched in the MG (macroglia) cluster (**panel c-d**). It was recently reported that *Mafb* contributes to the activation of microglia². It also identified *Hoxd9* enriched in IN (inhibitory neuron) from the imputed but not the raw data. Similarly, we found that *Dlx2*, *Lhx8*, *Arx* and *Neurog1* are much more significantly enriched in the respectively clusters in the imputed data (**panel c-d**). *Dlx2*, *Lhx8* and *Arx* are important components in the MGE (medial ganglionic eminence) pathway of forebrain development³, and *Neurog1* is required for excitatory neurons in the cerebral cortex⁴.

We discussed these findings in the text **(line 236, 248)**.

5. The authors discuss the interpretability of the SCALE features beginning on line 272,

however they do not discuss what the sets of chosen peaks are, for example, it would be interesting to see a gene set enrichment analysis of the nearest (or predicted) genes of the peaks that make up a feature. A similar analysis of the batch effect peaks would also be interesting.

RESPONSE: We thank the reviewer for the suggestion, which gives us the chance to present an in-depth discussion on the biological relevance of individual features.

We used the GREAT algorithm (version 3.0.0) for gene enrichment analysis, following *Preissl et al.*⁵: for each peak, we included genomic regions of a basal plus an extension (1kb upstream and 0.1kb downstream with up to 500-kb max extension) in the search for elements enriched with the GO 'biological process' terms. We found that features learned by SCALE are frequently disentangled and represent specific biological functions (line 318, 341, and the new **Supplementary Figure S16**).

For example, in the Leukemia dataset, feature 9-associated peaks were specific to the immune cell Monocyte, and were enriched in regulatory elements of immune-related biological process (BP, Methods) (**panel b**). In the Forebrain dataset, feature 3 was linked to the AC (astrocyte) and the OC (oligodendrocyte) cell types, and enriched in elements related to "glial cell differentiation" (**panel c**). In the Splenocyte dataset, features 4 and 7 respectively characterized two close-to-complementary sets of cell types (**panel d**), with feature 4 more enriched in B cells and related biological process like "myeloid leukocyte activation and differentiation", and feature 7 more of T cells with terms such as "immune response" and "regulation of cell killing".

We also checked the batch effect-related features of the Breast Tumor dataset. Most of plate-related features are not associated with any biological meaning, except for feature 8, which was associated with one plate of CD45+ cells, and enriched in biological process such as "response to cytokine stimulus". We communicated with Dr. Xingqi Chen on this observation, and Xingqi pointed out that this might be because the two separate plates contained cells of different cell states (**personal communication**). We analyzed the distributions of CD45 FACS signal in this dataset, and found that indeed there exists a little difference in plate 3 and 4, although they are both CD45+ (p-value: 0.061, Mann Whitney U-test) (the following **Revision Figure R1**)

Revision Figure R1. CD45 FACS Signal of four plates.

6. Newer scATAC-seq datasets often have median reads per cell in the 10s of thousands. The authors use one dataset of similar density (splenocyte), but I would be interested in seeing another example, especially as the splenocyte dataset most often had similar performance with scVI. An example of denser datasets available are those published by 10X Genomics from their new platform.

RESPONSE: We have now applied SCALE on 10 denser datasets from 10X Genomics (Single Cell ATAC Demonstration, Cell Ranger ATAC 1.1.0) (<https://support.10xgenomics.com/single-cell-atac/datasets>). It should be noted that these datasets do not provide cell type information with biological definition, i.e., those cell labels were generated from clusters computationally inferred by graph clustering. But in order to compare different tools, we still used these labels as “reference”.

The following **Revision Figure R2** shows data embedding colored by the reference clusters, the predicted clusters and the confusion matrices by SCALE, scVI and cisTopic on these 10 datasets. As the figure shows, SCALE achieves very good performance on all tested datasets and the overall performance is superior to scVI and cisTopic. However, because the “reference” clusters in these datasets are only computationally inferred with no biological definition, we decided not to include the results in the manuscript. In addition, the figure is very large but we thought it does not add much information (i.e., the conclusions remain the same).

Revision Figure R2. Comparison of SCALE, scVI and cisTopic with reference embedding and clusters on 10X Genomics scATAC-seq datasets.

7. Figure S5: I am curious about the limits of the SCALE method to downsampling. The authors should continue to downsample the data beyond 0.3 (perhaps plotted using line charts rather than barcharts). It would also be useful to use downsampling to investigate the relationship between performance and reads per cell, for example by plotting the number of reads per cell after downsampling compared to performance.

RESPONSE: We thank the reviewer and agree that using line charts is better than barcharts. We have now changed to line charts.

We also followed the suggestion to extend the down-sampling to 0.9, and included the distribution of the number of peaks per cell (the new **Supplementary Figure S6**). SCALE displayed only a moderate decrease in performance until at about the corruption level of 0.6, and the overall clustering results are better than SC3 and scVI.

8. Was any tuning required to run SCALE on the different datasets provided? Were any non-default options used?

RESPONSE: SCALE required a little tuning in the originally submitted version, where we used the same parameters for all datasets, except for the learning rate. It was set to an initial value of 0.002 with a 10% decay in every 10 epochs until 0.0002 for the Leukemia, InSilico, GM12878vsHEK, GM12878vsHL datasets (these are small datasets), but set to a constant value of 0.0002 for the rest. This strategy was to accelerate the convergence of learning for small datasets. In the revision, we made a change and used the same strategy for any input data, i.e., a learning rate with decay by iteration instead of by epoch. The new approach worked very well and allowed similar, stable convergence of learning.

We now included a new **Supplementary Table S3**, which provides all details of the parameters.

9. Lastly, the authors should make some comment about the computational performance of the method – even just a note on whether it takes seconds, minutes, or hours to run on some moderately sized dataset.

RESPONSE: We thank the reviewer for his/her suggestion to evaluate the computational performance of SCALE. We now include a new **Supplementary Figure S21**, which shows the running time and memory usage benchmarked on datasets down-sampled

from the mouse atlas dataset (10,000 peaks and different number of cells). The running time and memory usage are very stable for different scales of datasets (line 372). SCALE requires a little over 1.5 hours and 2 GB of memory to run a moderately sized dataset.

Minor points:

Line 47: Cusanovich et al. Science 2015 should also be cited along with Buenrostro for the invention of single-cell ATAC-seq.

RESPONSE: We thank the reviewer for pointing this out. We have cited both Cusanovich et. al. and Buenrostro et. al. for the invention of single-cell ATAC-seq (line 46).

Line 138: Identifies cell types, does not discover them.

RESPONSE: We have followed the suggestion and corrected “identifies” to “discover” (line 140).

Figure S3: The authors should remove the bottom rows from the confusion matrices in rows 2 and 3.

RESPONSE: We have followed the suggestion and removed the bottom rows from the confusion matrices in rows 2 and 3 (the new **Supplementary Figure S4**). However, after including more data, we switched the rows and columns, so now it removal works for columns 2 and 3.

In several figures, the authors use a three-color scale (red to white to blue) when displaying values that do not have a natural mid-point (i.e. all are positive or negative, none pass zero – the white midpoint has no special meaning). The figures would be easier to decipher on a two-color scale.

RESPONSE: We totally agree with the suggestion and have changed the three-color scale to two-color scale (white to blue, the new **Supplementary Figures S5** and **S9**).

REVIEWER #2 (Remarks to the Author):

In this manuscript Lei Xiong et al. applied a variational autoencoder combining with Gaussian Mixture Model to encode latent features that could accurately characterize

single cell ATAC-seq data. scATAC-seq is binary, sparse and noisy. So it is difficult to directly apply analysis methods developed for bulk ATAC-seq data. Also methods developed for scRNA-seq cannot be directly applied to analyze scATAC-seq due to the nearly binary nature of the later. Lei Xiong applied variational encoder which learns a latent distribution of the observed data through an encoder (recognition model) and the decoder (generative model). However, regular VAE often underfits sparse data. So they introduced Gaussian Mixture Model (GMM) as a prior over the latent variables that gives more power in fitting sparse data than regular VAE. Overall this is an important work given how few methods/tools are available that supports scATAC-seq analysis. However, I feel that some more work is needed before I can recommend it for publication.

RESPONSE: We appreciate the reviewer for the enthusiastic assessment and thoughtful comments that have helped us to improve the manuscript, where important revised text, figures, tables are highlighted in yellow.

Comments:

1. Although the authors have benchmarked the tool with number of published datasets, the number of single cells were moderate (the maximum was 3166 cells). As, 10X and ICELL8 are now launching their scATAC-seq protocols it would soon be common to have scATAC-seq applied to large number of single cells. So I would like to see how the tool scales to large datasets. One such dataset could be of Cusanovich et. al. Cell (2018) DOI:<https://doi.org/10.1016/j.cell.2018.06.052> with ~100,000 single cells. Even if it is not possible to work with all the cells, it would be good to know how many cells SCALE can handle and how the time/memory scales with increasing number of cells.

Also, in that paper Cusanovich reported identification of 30 major clusters which were then broken down to 85 clusters in total. How many of these clusters can SCALE pick?

RESPONSE: We totally agree with the importance to test SCALE on large datasets. We have followed the suggestion and benchmarked the performance, including running time/memory of SCALE, on the mouse atlas dataset of Cusanovich et. al. We found the results to be satisfactory.

Cusanovich et. al. defined 30 cell types by computational clustering, which were used as “reference” cell types when we benchmark SCALE (please see the response to Question 6 of Reviewer #1 for more discussion). We used SCALE to also define 30 clusters and

calculated the confusion matrix comparing with the “reference” 30 clusters. As shown in the new **Supplementary Figure S20**, most major clusters have a corresponding one identified by SCALE. From the confusion matrix, we can see that there are 18 clusters that are consistent between SCALE and the “reference” clustering (F1 score greater than 0.3 in each cluster). The overall weighted average F1 score is 0.47, suggesting a good correspondence between SCALE and the “reference” clusters (**Revision Table RT1**). Nevertheless, some big clusters are split into two or three smaller groups by SCALE. It is worthy to note that the original “reference” clustering is not necessarily better than clustering by SCALE.

	precision	recall	f1-score	support
1	0.42	0.28	0.34	6642
2	0.96	0.43	0.6	6400
3	1	0.49	0.66	5664
4	0.95	0.49	0.65	5462
5	0.69	0.53	0.6	4890
6	0.53	0.32	0.4	4783
7	1	0.49	0.66	4197
8	0.93	0.65	0.77	4099
9	0.35	0.35	0.35	4048
10	0.34	0.32	0.33	3425
11	0.67	0.59	0.63	3164
12	0.74	0.5	0.6	3055
13	0.98	0.83	0.9	2811
14	0.24	0.49	0.32	2089
15	0.25	0.36	0.29	2071
16	0.08	0.18	0.11	1943
17	0.33	0.38	0.36	1931
18	0.25	0.18	0.21	1804
19	0.38	0.62	0.47	1666
20	0.32	0.41	0.36	1622
21	0.12	0.17	0.14	1558
22	0	0	0	1504
23	0.32	0.56	0.41	1319
24	0.26	0.3	0.28	1173
25	0.06	0.18	0.09	973
26	0.04	0.11	0.06	729
27	0	0	0	672
28	0.15	0.65	0.24	571
29	0	0	0	515
30	0	0	0	393
accuracy			0.42	81173
macro avg	0.41	0.36	0.36	81173
weighted avg	0.59	0.42	0.47	81173

Revision Table RT1. Clustering report of SCALE on mouse atlas dataset.

We also evaluated the computational performance with respect to the dataset scales. Please see our response to Question 9 of Reviewer #1, and the new **Supplementary Figure S21**, the running time and memory usage a set of benchmark datasets do not increase with the size of datasets (**line 372**).

2. It is mentioned in the paper that SCALE is insensivie to the encoder structure and the dimension of latent features. But no benchmarking results based on different structure and dimensions of latent spaces were presented. Presenting this data in the paper would be very useful.

RESPONSE: We have tested different combination of encoder structures (128, 1024-128,

1024-256-128, 1024-512-128, 3200-400, 3200-800-400, 3200-1600-800-400) with latent dimensions (8, 10, 12, 15, 20, 50). The results showed that SCALE is insensitive to the encoder structure and the dimension of latent features. Please refer to the new **Supplementary Table S1** for details.

3. It is always challenging to predict the number of clusters 'K' in the model. Users can run GMM separately and use BIC or other such measures to identify possible number of clusters. But an embedded approach with SCALE to predict the 'K' would be easier for the user to run SCALE all at one go.

RESPONSE: As suggested, we have run GMM separately and calculated BIC at different number of clusters. We also tested another method used by SC3 to estimate cluster number according to the Tracy-Widom distribution. As the following **Revision Figure R3**, **Revision Table RT2** show, estimations based on the Tracy-Widom distribution, but not the BIC metric, are often close to the reference number of clusters.

We see that SCALE clustering at estimated numbers of clusters from the Tracy-Widom distribution also well corresponds to the reference clusters (**line 203**, the new **Supplementary Figure S7**).

We thank the reviewer for the nice suggestion to provide an estimation of the number of clusters in SCALE. We have implemented the idea based on the Tracy-Widom distribution and users can now get clustering results with the estimated number of clusters 'K' if the parameter ('K') is not specified.

Revision Figure R3. BIC scores under different cluster number of six datasets.

	Leukemia	GM12878vsHEK	GM12878vsHL	InSilico	Splenocyte	Forebrain
ref k	6	2	2	6	12	8
estimated k (BIC)	2	2	2	2	2	2
estimated k (Tracy-Widom)	5	2	2	8	17	12

Revision Table RT2. Best k estimated by BIC methods and Tracy–Widom methods.

4. When there would be minor clusters along with major clusters, for eg. with Cusanovich dataset, which one should be used for the GMM?

RESPONSE: The major clusters should be used. It seems that GMM of SCALE cannot discover minor clusters along with major ones just by increasing the cluster number. For the example of the Cusanovich dataset (30 major clusters which can be broken down to 85 clusters), SCALE got similar embeddings with the cluster number of 85 as those with the cluster number of 30 (compare Revision Figure R4 and Supplementary Figure S20).

Revision Figure R4. t-SNE of SCALE features (k=85) colored by 85 minor clusters.

5. The imputed data from SCALE is floating point values, but the scATAC-seq is effectively binary (counts of more than one would represent other alleles of a locus, but this would be rare) value. Downstream algorithms that applies binary assumption of the data would not work on this imputed data. How would the user address this issue? Also, the author mentions in the paper that SCALE recovered the original peak profiles in the process of data regularization by GMM. But will it not be the case that the original peak profiles would be of binary value?

RESPONSE: We thank the reviewer for pointing out this issue, which gives the opportunity to further improve the manuscript and the software. Actually the float imputed values can be interpreted as the “access potential” of the peak region in the cell and is

compatible with chromVAR. We now implemented the binarization of imputed data and compared the raw data with the binarized and non-binarized imputed data on cluster-specific peaks (**Methods: Binarization**). As the new **Supplementary Figure S10** shows, imputation with and without binarization could both improve the enrichment of cluster-specific peaks.

We have now included the binarization of the imputed data as a post- data processing option when running SCALE (**option: *--binary***).

It worth noting that our GMM regularization is based on the cell distribution not on peak distribution, so it doesn't matter whether peak profiles are binary value or float value.

6. The authors showed that SCALE could identify features that are related to batch effects from the input data in their Breast Tumor dataset. In the GM12878 cell line data, PCA analysis was separating cells based on replicates. But the figure they are referring to are t-SNE plots and I could not find how many PCAs were taken as input to generate this t-SNE plot. Also, which Principle Component was responsible for the batch effect.

SCALE could not detect any batch effect in GM12878 cell line dataset. The authors argued that the batch effect might not be strong enough to come up as a latent space feature and thus was not detected as a latent space. But this seems to be ambiguous to me. If SCALE has the capability of identifying batch effect it should do so even when the batch effect signal is not strong. The user can then work around this feature to take in to consideration for further downstream analysis.

RESPONSE: We especially thank the reviewer for urging us to investigate the issue that SCALE did not identify any batch effect in the GM12878 cell line. First, we will clarify some technical details. We used the first ten components of PCAs to generate this t-SNE plot. From the heatmap of ten-components of PCAs, we could say that components 2, 3 and 5 are the most responsible for the batch effect (the new **Supplementary Figure S17**).

When revisiting the analysis, we noticed that the GM12878 dataset is somewhat different from other scATAC-seq datasets, whose peak values are quantitative with many greater than 2, and some are of really high values (>1000. See the following **Revision Figure R5**). These relatively large values caused variations that affect PCA analysis. However, when we tried to remove these large values by binarization, the batch effect disappeared

in the PCA analysis. We checked the top 200 specific peaks for each replicate and barely found any replicate-related peaks, which further confirmed the observation. We speculate that, as a linear method, PCA suffers from large data variation in quantitative differences. On the other hand, SCALE seems insensitive to it since SCALE mostly performs non-linear transformations such as sigmoid.

Revision Figure R5. Distribution of peak number with value > 2 in the count matrix across different replicates (left). Distribution of peak values with value > 2 in the count matrix of the combined GM12878 dataset (right).

7. What specific attributes of the dataset causes this differences in identifying batch effect, is it the sparsity of the data or heterogeneity in the data or something else?

RESPONSE: We think that it is the data heterogeneity that causes this difference. Taking the Mouse Breast Tumor dataset as an example, first, we were able to identify 1000 specific peaks for each plate in the raw data (**Revision Figure R6a**). The existence of these plate-related peaks suggests some level of “batch-effect” between the plates of same biological cell types (plate 1 vs plate 2, plate 3 vs plate 4). These peaks remain to be plate-specific after data imputation, which has greatly reduced data sparsity (**Revision Figure R6b**).

Revision Figure R6. Top 1000 specific peaks of each plate of the raw (a) and the imputed data (b) of the Mouse Breast Tumor dataset.

8. I tried the tool in one of our local dataset where the data for the peaks has been binarized (any value more than 1 has been made 1 although they were few). In the first Epoch it gave a loss value and lr, but after that all the epoch produced nan and ended with error. This needs to be fixed.

RESPONSE: We highly appreciate the reviewer for testing our software and pointing out the issue. Some of the reasons for this problem could be:

a, incompatible Python versions. We only support Python 3.6+ for the moment. Please make sure that the version of Python is newer than 3.6.

b. the input data may contain some low-quality cells or peaks. One solution is to clean the input data before running SCALE. We now have provided a data-filtering option to remove peaks that appear in less than x% all cells, and also cells that have peaks with signals less than certain number or ratio of total number of peaks.

c. a previous bug in model learning. In the originally submitted version of SCALE, the learning rate was initialized to 0.002 with a decay of 10% every 10 epochs until 0.0002. This strategy is defined to speed up the training of small datasets. However, it may fail on large datasets, where the learning must go through many iterations (in each iteration a

small batch of 32 samples fed to the model) before it is reduced to a stable value 0.0002. However, large learning rates will lead to weight overflow which may cause *nan* values. In the new version of SCALE, we solved this problem by decaying learning rate along iterations instead of epochs.

Finally, we are happy to help testing the dataset if the reviewer can upload the data onto some locations.

Reviewer #3 (Remarks to the Author):

This paper proposes to use variational autoencoders (VAE) with a Gaussian mixture model (GMM) latent prior distribution for analysis of scATAC-seq data (Single-cell profiles of chromatin-accessibility landscape at single cellular level). This is a timely contribution, well-written with a thorough analysis of a number of datasets. A number of comments focussing on the modelling part are given in the following:

RESPONSE: We appreciate the reviewer for the enthusiastic assessment. We have followed the comments and improved the manuscript, which are highlighted in yellow.

Major comments:

1. The Gaussian mixture model (GMM) variational autoencoder (VAE) model has been applied to scRNA-Seq previously <https://www.biorxiv.org/content/10.1101/318295v2> This does not invalidate this paper because its main focus is ATAC-seq but the paper is relevant to cite.

RESPONSE: We thank the reviewer for the reminder. We have added the reference (line 88).

2. K-means clustering is applied to clustering of the learned latent features. Why not use the responsibilities (soft cluster assignments) that the algorithm gives. The GMM is a clustering algorithm by itself.

RESPONSE: We agree that we could use the responsibilities of GMM for clustering. Actually, we found that the soft assignment of GMM is generally good. However it is not

as robust and accurate as k-means clustering using the latent features. We found that usually the GMM responsibilities would adapt to the cluster center. However, the component centers are not always consistent with the embedding of latent representations. A small bias of components center may lead to a group of cells assigned to the wrong cluster.

3. The ATAC data is sparse from missing data as stated in the paper: "Normally, only a few thousand distinct reads (versus many thousands of possible open positions) are obtained per cell, thus resulting in many bona fide open chromatin sites of the cell without sequencing data signals (i.e., peaks)." It could be good with a discussion about what is the adequate model of missingness and how that affects the formulation of the generative model. It could be the one proposed. But more justification is needed.

RESPONSE: We followed the suggestion and tested the impact of missingness on the generative model by whether the imputed data can restore the original data structure (inter and intra-correlation of the meta-cells). As the new **Supplementary Figure S13** shows, for all test datasets, when the corruption level went beyond 0.5, the generative model started to miss original data structure, where cells from different clusters started to become similar to each other and deviate from the original meta-cells. But please note that here the data corruption is performed on the basis of real test datasets, which already contains a high degree of data loss. In other words, here a corruption level of 0.5 represents a very severe data loss.

Minor comments:

1. The analysis of scATACseq data hence suffers from the curse of sparsity.. Perhaps it is more accurate to say

The analysis of scATACseq data hence suffers from the curse of missingness

RESPONSE: We agree and have corrected as suggested **(line 52)**.

2. The Seurat method <https://satijalab.org/seurat/> seems to emerge as a standard method for scRNA-Seq analysis. Could be good to see a comparison to Seurat.

RESPONSE: We thank the reviewer for the reminder. We now include a comparison to

Seurat in the revised manuscript, following the tutorial at https://satijalab.org/seurat/v3.0/pbmc3k_tutorial.html. For consistency, we did not filter cells. **Revision Figure R7** shows the results of embedding and clustering. As we can see, the performance of Seurat is similar to other scRNA-seq tools like SC3 and scVI, i.e., not very good for scATAC-seq data analysis. We choose to not include the details of this comparison in the revised manuscript.

Revision Figure R7. Confusion matrix and tSNE embedding of Seurat results.

3. Good to see the code is available!

RESPONSE: Thank you to the reviewer. We will keep updating the code (<https://github.com/jsxlei/SCALE>) to improve the tool. We hope that SCALE will help scATAC-seq data analysis.

- 1 Cusanovich, D. A. *et al.* A Single-Cell Atlas of In Vivo Mammalian Chromatin Accessibility. *Cell*, doi:10.1016/j.cell.2018.06.052 (2018).
- 2 Tozaki-Saitoh, H. *et al.* Transcription factor MafB contributes to the activation of spinal microglia underlying neuropathic pain development. *Glia***67**, 729-740, doi:10.1002/glia.23570 (2019).
- 3 Nord, A. S., Pattabiraman, K., Visel, A. & Rubenstein, J. L. R. Genomic perspectives of transcriptional regulation in forebrain development. *Neuron***85**, 27-47, doi:10.1016/j.neuron.2014.11.011 (2015).
- 4 Kim, E. J. *et al.* Spatiotemporal fate map of neurogenin1 (Neurog1) lineages in the mouse central nervous system. *J Comp Neurol***519**, 1355-1370, doi:10.1002/cne.22574 (2011).
- 5 Preissl, S. *et al.* Single-nucleus analysis of accessible chromatin in developing mouse forebrain reveals cell-type-specific transcriptional regulation. *Nat Neurosci***21**, 432-439, doi:10.1038/s41593-018-0079-3 (2018).

REVIEWERS' COMMENTS:

Reviewer #1 (Remarks to the Author):

The authors have addressed all of my previous concerns and demonstrated performance in comparison to several new methods. I recommend this manuscript for publication.

Reviewer #2 (Remarks to the Author):

The authors have addressed my concerns satisfactorily.

Syed Murtuza Baker

Reviewer #3 (Remarks to the Author):

The reviews were very consistent in their independent feedback and the authors have made a serious effort to address the points raised. This taken together with the fact that the first version was already in a very good shape I can recommend publication without further comments.

We appreciate all three reviewers for spending much time reviewing our paper and helping us improve the work.